# Pulmonary complications observed in patients with infective endocarditis with and without injection drug use: An analysis of the National Inpatient Sample

**Jenny T. Bui**[1], **Asher J. Schranz**[2], **Paula D. Strassle**[3], **Chris B. Agala**[3], **Gita N. Mody**[4], **John S. Ikonomidis**[4], **Jason M. Long**[4]*

1 University of North Carolina at Chapel Hill School of Medicine, Chapel Hill, North Carolina, United States of America, 2 Department of Medicine, Division of Infectious Disease, University of North Carolina at Chapel Hill, Chapel Hill, North Carolina, United States of America, 3 Department of Surgery, University of North Carolina at Chapel Hill, Chapel Hill, North Carolina, United States of America, 4 Department of Surgery, Division of Cardiothoracic Surgery, University of North Carolina at Chapel Hill, Chapel Hill, North Carlina, United States of America

* jason_long@med.unc.edu

**Data Availability Statement:** The data used for this analysis are publicly available for purchase through

## Abstract

### Background

The impact of cardiovascular and neurologic complications on infective endocarditis (IE) are well studied, yet the prevalence and significance of pulmonary complications in IE is not defined. To better characterize the multifaceted nature of IE management, we aimed to describe the occurrence and significance of pulmonary complications in IE, including among persons with IE related to drug use.

### Methods

Hospitalizations of adult ($\geq$18 years old) patients diagnosed with IE were identified in the 2016 National Inpatient Sample using ICD-10 codes. Multivariable logistic and linear regression were used to compare IE patient outcomes between those with and without pulmonary complications and to identify predictors of pulmonary complications. Interaction terms were used to assess the impact of drug-use IE (DU-IE) and pulmonary complications on inpatient outcomes.

### Results

In 2016, there were an estimated 88,995 hospitalizations of patients diagnosed with IE. Of these hospitalizations, 15,490 (17%) were drug-use related. Drug-use IE (DU-IE) had the highest odds of pulmonary complications (OR 2.97, 95% CI 2.50, 3.45). At least one pulmonary complication was identified in 6,580 (7%) of IE patients. DU-IE hospitalizations were more likely to have a diagnosis of pyothorax (3% vs. 1%, p<0.001), lung abscess (3% vs. <1%, p<0.001), and septic pulmonary embolism (27% vs. 2%, p<0.001). Pulmonary complications were associated with longer average lengths of stay (CIE 7.22 days 95% CI 6.11,

the Healthcare Cost and Utilization Project (HCUP) at the National Inpatient Sample website (https://www.hcup-us.ahrq.gov/nisoverview.jsp) though we are not permitted to make the data publicly available for free per the data use agreement with HCUP. Data were purchased publicly from HCUP and made available for author use via secure digital download. All interested researchers can access the data through HCUP in a similar manner. None of the authors of this manuscript are affiliated with HCUP or are data owners or collected any of these data on behalf of HCUP or data owners.

**Funding:** AJS was funded by the National Institute on Drug Abuse (K23DA049946) and the National Institute of Allergy and Infectious Diseases (T32AI070114). The funders had no role in study design, data collection and analysis, decision to publish, or preparation of the manuscript.

**Competing interests:** The authors have declared that no competing interests exist.

8.32), higher hospital charges (CIE 78.51 thousand dollars 95% CI 57.44, 99.57), more frequent post-discharge transfers (acute care: OR 1.37, 95% CI 1.09, 1.71; long-term care: OR 2.19, 95% CI 1.83, 2.61), and increased odds of inpatient mortality (OR 1.81 95% CI 1.39, 2.35).

## Conclusion and relevance

IE with pulmonary complications is associated with worse outcomes. Patients with DU-IE have a particularly high prevalence of pulmonary complications that may require timely thoracic surgical intervention, likely owing to right-sided valve involvement. More research is needed to determine optimal management strategies for complications to improve patient outcomes.

## Introduction

Infective endocarditis (IE) is a severe infection of one or more heart valves that confers a high rate of morbidity and mortality if left untreated [1]. A common risk factor of IE is injection drug use that can introduce bacteria or fungi into the bloodstream, and ultimately cause vegetation and/or abscesses of the heart valves that often requires surgical intervention [1].

Drug use-related IE (DU-IE) primarily affects the tricuspid valve. As a result, vegetation and septic emboli are introduced into the pulmonary circulation. This in turn can lead to pulmonary complications, such as empyema, septic pulmonary emboli, and lung abscesses (Figs 1 and 2). At our institution, we have observed an increase in both DU-IE and the incidence of pulmonary complications associated with IE (IE-PC) requiring general thoracic surgical consultation and intervention. Nearly one-third of our IE patients have required thoracic surgical intervention, such as pulmonary decortications, lung resections, and/or percutaneous drainage of the pleural space between 2015 and 2019 (pending publication).

The impact of cardiovascular complications, such as valvular damage, cardiac abscess, conduction abnormalities, and stroke on IE outcomes are well-studied [1–5]. However, a recent literature search regarding the effect and management of pulmonary complications (empyema, lung abscesses, and septic pulmonary emboli) on outcomes of IE consists mainly of case reports [6–15]. There are no large scale epidemiological, retrospective, or prospective studies on the subject and the epidemiology of IE-PC is largely unknown. Further, the optimal timing of intervention for pulmonary complications and the need for cardiac surgical intervention for cardiac complications (damaged/insufficient valves, cardiac abscess/perforation) remains unknown. Therefore, we sought to assess the following: (1) determine the incidence of pulmonary complications in IE; (2) identify predictors of pulmonary complications in patients with IE; (3) characterize the short-term outcomes including length of stay (LOS), hospital charges, and discharge disposition (including inpatient mortality) in IE patients with pulmonary complications with and without drug use.

## Materials and methods

### Data source and patient population

Between July and November 2019, we identified all patients admitted for IE between January 1, 2016 and December 31, 2016 in the Healthcare Cost and Utilization Project National Inpatient Sample (NIS). We focused on 2016 in order to facilitate usage of International

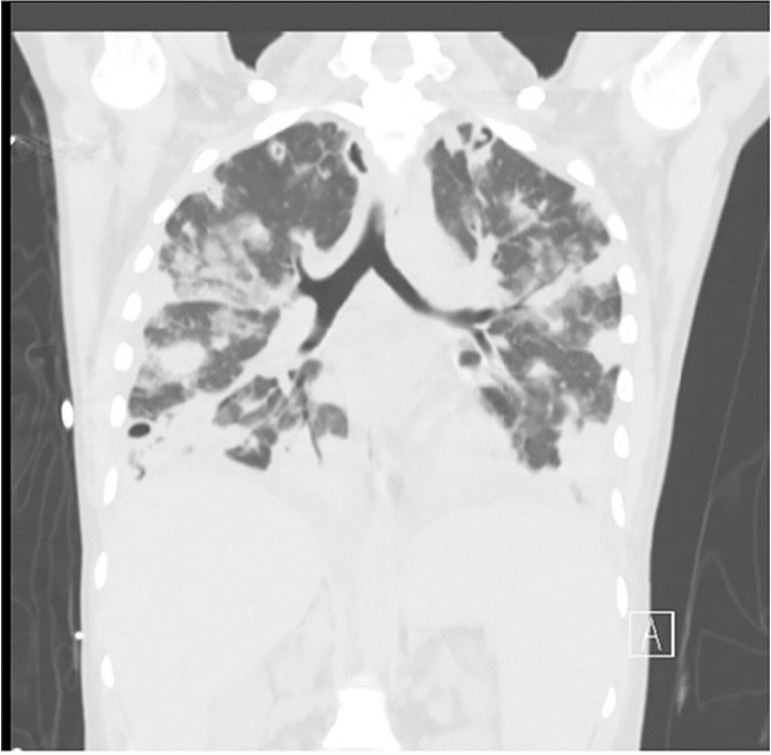

**Fig 1. Diffuse septic pulmonary emboli.** Coronal computed tomography scan depicting diffuse septic pulmonary emboli in a patient with tricuspid valve endocarditis and history of injection drug use.

Classification of Disease, 10th edition, clinical modification (ICD-10-CM) codes, which had been implemented in late 2015, and NIS 2017 was not available at the time of the study analysis. The NIS is a publicly available all-payer database in the United States and includes roughly 7 million hospitalizations (35 million weighted hospitalizations) from community hospitals in the US. Discharge weights were used to obtain national estimates, and appropriate methods were used to account for the complex sample design. Discharge weights were calculated and provided by HCUP and account for changes in the stratified sampling approach. Sampling strata include Census division, location (urban vs. rural), teaching status, hospital control (i.e. government or private), and bed size.

### Inclusion and exclusion criteria

Hospitalizations of adult (≥18 years old) patients diagnosed with IE were identified using ICD-10-CM codes and included in the study population, S1 Table in S1 File [16–20]. Patients discharged against medical advice (n = 768, 4%, unweighted) or with unknown disposition (n = 90, <1%, unweighted) were excluded since their course of care and clinical outcomes could not be reliably assessed. IE-PC of interest included pyothorax (used in place of empyema as empyema does not have a diagnosis code), lung abscess, and septic pulmonary embolism (S1 Table in S1 File). Drug use was also identified using ICD-10-CM diagnosis codes (S1 Table in S1 File). Drug use was defined as either a) the presence of a code for abuse, dependence or toxicity of a drug prone to injection or B) a diagnosis of Hepatitis C in adults ≤50 years old [16]. Only secondary diagnosis codes (i.e. DX2 –DX30) were used to capture complications.

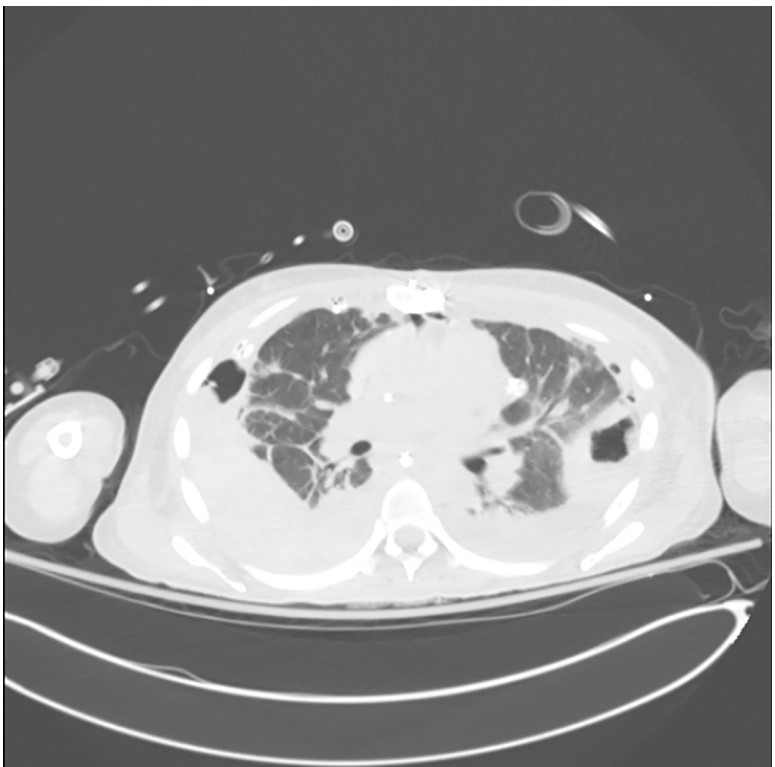

**Fig 2. Bilateral cavitary lesions.** Axial computed tomography scan depicting bilateral cavity lung lesions consistent with lung abscesses and empyema in a patient with tricuspid valve endocarditis and history of injection drug use.

## Outcomes of interest

The primary outcome of interest, as specified in our aims, was inpatient mortality in IE-PC patients (with and without drug use). The secondary outcomes of interest included predictors of pulmonary complications, undergoing a thoracic and/or heart valve procedure (identified through ICD-10-CM procedure codes, S2 Table in S1 File), discharge disposition, LOS, total hospital charges, and time to thoracic or heart valve procedure (among those who underwent procedure only).

## Statistical analyses

Multivariable generalized logistic, logistic and linear regression were used to compare IE patient outcomes between those with and without pulmonary complications (pyothorax, lung abscess, septic pulmonary embolism). Multivariable generalized logistic regression was also used to identify the predictors of having any pulmonary complication. All models were adjusted for patient age, gender, race/ethnicity, primary insurance, median income in the patient's ZIP code, Charlson Comorbidity Index (CCI), drug use, hospital bed size, teaching status, region, ownership, elective admission, ED status, transfer status, and presence of a cerebral complication. Age and CCI were modeled as restricted quadratic splines [21]. We opted to not highlight specific comorbidities given the likely low prevalence of these comorbidities in our patient population and the potential that many multisystem comorbidities may impact outcomes in patients with endocarditis or those undergoing surgery. We instead relied on the CCI as a summative measure. CCI was calculated using codes identified through forward and backward mapping of Deyo et al. algorithms using the CMS Generalized Equivalence

Mappings. These codes have been used previously [22]. Cerebral complications (abscess, hemorrhage, infarction, ischemia) were identified using ICD-10-CM codes G06.0, I60.00-I60.9, I61.0-I63.9, and I67.82. Appropriate survey-specific analysis methods were used to account for clustering and stratification and weighted to obtain national estimates. Potential effect measure modification by drug use was also assessed. Interaction terms between pulmonary complication and drug use were added to the models above to obtain effect estimates for IE with and without drug use. To gauge potential effect of large sample size on multivariate estimates of group comparisons, secondary inferential analyses were conducted to assess crude associations between outcomes and hospital regions.

All analyses were performed using SAS 9.4 (SAS Inc., Cary, NC). All reported results are weighted unless otherwise noted.

## Results

### Patient characteristics

In 2016, there were an estimated 88,995 hospitalizations of patients diagnosed with IE—the majority of which were non-elective, non-transfer hospital admissions (Table 1). Within the IE hospitalized cohort, 15,490 (17%) were found to be DU-IE. DU-IE was most common among adults between 30–39 years old (4,710, 30%) and least common among adults between 65–79 years old (605, 4%). The most common drug associated with DU-IE was opioids (inclusive of opiates), reported in 64% of the DU-IE population. Other drugs (without concurrent opioid use), including benzodiazepines, amphetamines or stimulants, and hallucinogens, were reported in 22% of the DU-IE population. Cocaine and Hepatitis C were the only drug-related diagnoses in 6% and 8% of this population, respectively. Among all IE patients, 6,580 (7%) had at least one pulmonary complication: 5,895 had a septic pulmonary embolism, 845 had an empyema, and 720 had a lung abscess. Of those with pulmonary complications, the median age was 34, 48% were male, 81% were non-Hispanic white, 67% used drugs and the median CCI score was 0.

### Predictors of pulmonary complications

Overall, DU-IE patients had the highest odds of pulmonary complications (OR 2.97, 95% CI 2.50, 3.45) of all variables analyzed, S3 Table in S1 File. Other predictors of pulmonary complications included elective admission to the hospital (OR 1.46, 95% CI 1.10, 1.94), acute transfer status (OR 1.76, 95% CI 1.40, 2.21), self-paying status (OR 1.36, 95% CI 1.08, 1.72), and treatment at an urban institution (non-teaching: OR 1.48, 95% CI 1.06, 2.07; teaching: OR 1.58, 95% CI 1.16, 2.15). A one-year increase in age was associated with a 5% decrease in the odds of having any pulmonary complication (OR 0.95, 95% CI 0.95, 0.95). Similarly, a one point increase in CCI was associated with a 16% decrease in the odds of pulmonary complications (OR: 0.84, 95% CI 0.80, 0.88). Compared to non-Hispanic (NH) White patients, NH Black and Hispanic patients had lower odds of pulmonary complications (OR 0.75, 95% 0.59, 0.96; OR: 0.71, 95% CI 0.54, 0.95). A similar result was also observed with the presence of a cerebral complication (OR 0.58, 95% CI 0.45, 0.76).

### Pulmonary complications and patient outcomes

Overall, patients hospitalized with IE-PC had an increased odds of inpatient mortality (OR 1.81, 95% CI 1.39, 2.35), compared to those without pulmonary involvement. IE-PC patients were also associated with, on average, a 7-day longer LOS (change in estimate [CIE] 7.22 days, 95% CI 6.11, 8.32), higher hospital charges (CIE 78.51 thousand dollars, 95% CI 57.44, 99.57),

**Table 1. Patient demographics and hospital characteristics, stratified by inpatient pulmonary complication.**

|  | Pulmonary Complication[a] 6,580 (7%) | No Pulmonary Complication 82,415 (93%) | p-value |
|---|---|---|---|
| **Age, years, median (IQR)** | 34 (27–46) | 66 (52–78) | <0.0001 |
| **Male, n (%)** | 3,145 (48) | 44,630 (54) | <0.0001 |
| **Race/ethnicity, n (%)** |  |  |  |
| Non-Hispanic White | 5,025 (81) | 57,115 (72) | <0.0001 |
| Non-Hispanic Black | 560 (9) | 10,725 (14) |  |
| Hispanic | 420 (7) | 7,090 (9) |  |
| Other[b] | 195 (3) | 4,065 (5) |  |
| *Missing* | 380 | 3,420 |  |
| **Primary insurance, n (%)** |  |  |  |
| Medicare/Medicaid | 4,275 (66) | 63,330 (77) | <0.0001 |
| Private | 1,070 (17) | 13,990 (17) |  |
| Self-pay/other | 1,120 (17) | 4,690 (6) |  |
| **Median ZIP code income[c], n (%)** |  |  |  |
| <$43,000 | 2,620 (41) | 26,735 (33) |  |
| $43,000-$53,999 | 1,725 (27) | 21,115 (26) | <0.0001 |
| $54,000-$70,999 | 1,230 (19) | 17,630 (22) |  |
| ≥$71,000 | 875 (13) | 15,205 (19) |  |
| **CCI, median (IQR)** | 0 (0–1) | 2 (0–3) | <0.0001 |
| **IV drug use, n (%)[d]** | 4,395 (67) | 11,095 (13) | <0.0001 |
| Opioid | 3165 (72) | 6670 (60) |  |
| Cocaine | 145 (3) | 855 (8) |  |
| Other drugs[e] | 760 (18) | 2600 (23) |  |
| Hepatitis C only | 325 (7) | 970 (9) |  |
| **Hospital bed size, n (%)** |  |  |  |
| Small | 930 (14) | 13,625 (17) | 0.0030 |
| Medium | 1,625 (25) | 23,305 (28) |  |
| Large | 4,025 (61) | 45,485 (55) |  |
| **Hospital location/type, n (%)** |  |  |  |
| Rural, non-teaching | 375 (6) | 7,575 (9) | <0.0001 |
| Urban, non-teaching | 1,325 (20) | 20,195 (25) |  |
| Urban, teaching | 4,880 (74) | 54,645 (66) |  |
| **Hospital region, n (%)** |  |  |  |
| Northeast | 1,030 (16) | 15,650 (19) | 0.0750[g] |
| Midwest | 1,325 (20) | 17,235 (21) |  |
| South | 3,015 (46) | 34,670 (42) |  |
| West | 1,210 (18) | 14,860 (18) |  |
| **Hospital ownership, n (%)** |  |  |  |
| Government, nonfederal | 1,000 (15) | 9,570 (12) | <0.0001 |
| Private, non-profit | 4,870 (74) | 59,370 (72) |  |
| Private, invest-own | 710 (11) | 13,475 (16) |  |
| **Elective admission, n (%)** | 375 (6) | 8,360 (10) | <0.0001 |
| **ED services, n (%)** | 4,170 (63) | 55,990 (68) | <0.0001 |
| **Admission source, n (%)** |  |  |  |
| Acute care transfer | 1,795 (28) | 11,680 (14) | <0.0001 |
| Other transfer | 220 (3) | 3,695 (5) |  |
| Non-transfer | 4,530 (69) | 66,720 (81) |  |

*(Continued)*

**Table 1.** (Continued)

| | Pulmonary Complication[a] 6,580 (7%) | No Pulmonary Complication 82,415 (93%) | p-value |
|---|---|---|---|
| **Cerebral complication, n (%)[f]** | 480 (7) | 8,780 (11) | <0.0001 |

Abbreviations: ED, emergency department; CCI, Charlson comorbidity index

[a] Complications included pyothorax, lung abscess, and septic pulmonary embolism; patients were able to have more than one complication

[b] Other race includes Asian/Pacific Islander, Native American, and 'Other' as categorized by HCUP; races were collapsed due to small cell sizes

[c] Quartile classification of the estimated median household income of residents in the patient's ZIP Code

[d] Drug use was categorized as opioid use (with or without other drugs), cocaine (with or without other non-opioid drugs), other drugs (no opioid or cocaine), and Hepatitis C only

[e] Other drugs include benzodiazepines, amphetamines or stimulants, and hallucinogens.

[f] Includes cerebral abscess, hemorrhage, infarction, and ischemia

[g] Results from secondary analyses showed some regional variations in all outcomes except heart valve surgery, even though the presence of pulmonary complications did not vary by hospital region (S3 Table in S1 File).

and more frequent post-discharge transfers (acute care: OR 1.37, 95% CI 1.09, 1.71; long-term care: OR 2.19, 95% CI 1.83, 2.61), Table 2. However, there was no difference in the likelihood of whether IE patients with or without pulmonary complications underwent a heart valve operation (OR 0.81, 95% CI: 0.63, 1.05). When a heart valve operation was performed, IE-PC patients were delayed by an average of 3 days compared to IE patients without pulmonary complications (CIE 2.83 days, 95% CI 1.54, 4.13).

The secondary inferential analyses showed some regional variations in all outcomes except heart valve surgery, even though the presence of pulmonary complications did not vary by

**Table 2.** Association of pulmonary complications and inpatient outcomes.

| | Pulmonary Complication | No Pulmonary Complication | | |
|---|---|---|---|---|
| | N (%) | N (%) | OR (95% CI)[a] | p-value |
| **Discharge disposition** | | | | |
| Routine | 2,155 (33) | 25,125 (30) | 1.0 (ref) | – |
| Home health care | 675 (10) | 15,720 (19) | 1.10 (0.89, 1.37) | 0.37 |
| Transfer, acute care | 875 (13) | 7,705 (9) | 1.37 (1.09, 1.71) | 0.006 |
| Transfer, long-term care | 2,330 (35) | 26,625 (32) | 2.19 (1.83, 2.61) | <0.0001 |
| Died | 545 (8) | 7,240 (9) | 1.81 (1.39, 2.35) | <0.0001 |
| **Inpatient procedure** | | | | |
| Thoracic procedure | 1,545 (23) | 3,365 (4) | 6.14 (5.01, 7.53) | <0.0001 |
| Heart valve surgery | 875 (13) | 6,610 (8) | 0.99 (0.79, 1.26) | 0.96 |
| Valve replacement | 650 (10) | 6,010 (7) | 0.81 (0.63, 1.05) | 0.11 |
| Other | 225 (3) | 70 (1) | 2.41 (1.57, 3.69) | <0.0001 |
| | Median (IQR) | Median (IQR) | CIE (95% CI)[a] | p-value |
| **Length of stay, days** | 15 (7–29) | 6 (3–12) | 7.22 (6.11, 8.32) | <0.0001 |
| **Hospital charges, thousands** | 129 (61–275) | 61 (29–137) | 78.51 (57.44, 99.57) | <0.0001 |
| **Thoracic procedure[b], days** | 7 (3–14) | 5 (2–12) | 0.69 (0.31, 1.07) | 0.0004 |
| **Heart valve surgery[b], days** | 8 (4–14) | 5 (1–10) | 2.83 (1.54, 4.13) | <0.0001 |

Abbreviations: OR, odds ratio; CI, confidence interval; IQR; interquartile range; CIE, change in estimate

[a] Adjusted for age, sex, race/ethnicity, primary insurance, median income in the patient's ZIP code, Charlson Comorbidity Index (CCI), IV drug use, hospital bed size, teaching status, region, ownership, elective admission, ED status, transfer status, and presence of cerebral complication; complex survey design and weighting were accounted for in analysis; age and CCI were modeled as restricted quadratic splines

[b] Among patients who underwent the procedure only

hospital region (S4 Table in S1 File). These results demonstrate that the differences seen in the outcomes of interest, such as length of stay, hospital charges, and discharge disposition (including inpatient mortality), are not only due to the differences seen in the baseline characteristics of our study population or a large sample size.

### Surgical order and patient outcomes

Only 995 of the IE hospitalizations underwent both a thoracic procedure and heart valve operation: 235 had a listed diagnosis of IE-PC and 760 did not. The majority of thoracic procedures were pleural drainages (4,520, 92%). Only seven percent of patients had operative pulmonary decortications (n = 365), while the remaining 1% had a lung resection (n = 25). Eighty-nine percent of patients who underwent a heart valve operation had a valve replacement (n = 6,660). In this patient subset, 270 (27%) underwent thoracic and cardiac procedures concurrently, 165 (17%) had the thoracic procedure first, and 560 (56%) had the heart valve surgery first. Thirty-five percent of those undergoing both thoracic and valve surgeries had DU-IE (n = 355). After adjustment, compared to those undergoing concurrent surgery, patients undergoing either the thoracic procedure or heart valve surgery first had a longer average LOS (CIE 6.28, 95% CI 2.87, 9.69 and CIE 4.46, 95% CI 1.28, 7.64, respectively). Patients undergoing thoracic procedures first also had higher hospital charges (CIE 91.3 thousand dollars, 95% CI 35.1,147.6). The surgical order of procedures did not result in a significant difference in mortality (6% of patients died who underwent heart valve surgery first compared to 3% for those who underwent a thoracic procedure first and 5% who underwent concurrent surgery).

### Pulmonary complications and drug use on patient outcomes

There was an estimated 15,490 hospitalizations of patients diagnosed with DU-IE (Table 3). Drug use-related hospitalizations more commonly had a diagnosis of pyothorax (3% vs. 1%, p<0.001), lung abscess (3% vs. <1%, p<0.001), and septic pulmonary embolism (27% vs. 2%, p<0.001). Even after accounting for patient characteristics, hospital characteristics, and cerebral complications (cerebral complications occurred 10% of both DU-IE (n = 1615) and non-DU-IE patients (n = 7645)), drug-related hospitalizations had greater odds of pulmonary complications (OR: 2.80, 95% CI 2.39, 3.30). They also demonstrated a trend of being more likely to undergo a thoracic procedure (OR 1.23, 95% CI 0.99, 1.52) or heart valve surgery (OR: 1.17, 95% CI: 0.98, 1.39), Table 4.

While pulmonary complications appeared to increase post-discharge transfers, inpatient mortality and hospital charges in both patients with and without DU-IE, the magnitude of the association between pulmonary complications and discharge disposition was more substantial in patients with DU-IE, compared to those with non-DU-IE (S5 Table in S1 File). For example, in patients with non-DU-IE, pulmonary complications were associated with a 48% increase in the odds of inpatient mortality (OR 1.48, 95% CI 1.05, 2.10), but among patients with DU-IE a pulmonary complication was associated with a 160% increase in the odds of inpatient mortality (OR 2.60, 95% CI 1.72, 3.83), p = 0.04.

## Discussion

To the best of our knowledge, this is the first analysis to describe the incidence and short-term outcomes of IE in patients with pulmonary complications with and without drug use using a comprehensive national database. Our data demonstrates that IE patients who develop pulmonary complications had a longer LOS, higher hospital charges, more frequent post-discharge

**Table 3. Patient demographics and hospital characteristics, stratified by drug use.**

|  | Drug Use[a] 15,490 (17%) | No Drug Use 73,505 (83%) | p-value |
|---|---|---|---|
| **Age, years, median (IQR)** | 35.9 (28–59) | 68.4 (56–79) | <0.0001 |
| **Male, n (%)** | 7,840 (16) | 39,935 (84) | 0.0003 |
| **Race/ethnicity, n (%)** |  |  |  |
| Non-Hispanic White | 11,570 (78) | 50,570 (72) | <0.0001 |
| Non-Hispanic Black | 1,500 (10) | 9,785 (14) |  |
| Hispanic | 1,195 (8) | 6,315 (9) |  |
| Other[b] | 525 (4) | 3,735 (5) |  |
| *Missing* | 700 | 3,100 |  |
| **Primary insurance, n (%)** |  |  |  |
| Medicare/Medicaid | 10,895 (72) | 56,710 (77) | <0.0001 |
| Private | 1,955 (13) | 13,105 (18) |  |
| Self-pay/other | 2.370 (15) | 3,440 (5) |  |
| **Median ZIP code income[c], n (%)** |  |  |  |
| <$43,000 | 6.350 (42) | 23,005 (32) |  |
| $43,000-$53,999 | 3,840 (25) | 19,000 (26) | <0.0001 |
| $54,000-$70,999 | 2,915 (19) | 15,945 (22) |  |
| ≥$71,000 | 2,000 (13) | 14,080 (20) |  |
| **CCI, median (IQR)** | 0 (0–1) | 2 (0–4) | <0.0001 |
| **Hospital bed size, n (%)** |  |  |  |
| Small | 2,330 (15) | 14,555 (17) | 0.02 |
| Medium | 3,990 (26) | 24,930 (28) |  |
| Large | 9,170 (59) | 49,510 (55) |  |
| **Hospital location/type, n (%)** |  |  |  |
| Rural, non-teaching | 1,065 (7) | 6,885 (9) | 0.001 |
| Urban, non-teaching | 3,530 (23) | 17,990 (25) |  |
| Urban, teaching | 10,895 (70) | 48,630 (66) |  |
| **Hospital region, n (%)** |  |  |  |
| Northeast | 2,755 (18) | 13,925 (19) | <0.0001 |
| Midwest | 2,645 (17) | 15,915 (22) |  |
| South | 6,865 (44) | 30,820 (42) |  |
| West | 3,225 (21) | 12,845 (17) |  |
| **Hospital ownership, n (%)** |  |  |  |
| Government, nonfederal | 2,340 (15) | 8,230 (11) | <0.0001 |
| Private, non-profit | 11,065 (71) | 53.175 (72) |  |
| Private, invest-own | 2,085(14) | 12,100 (17) |  |
| **Elective admission, n (%)** | 905 (6) | 7,830 (11) | <0.0001 |
| **ED services, n (%)** | 10,805 (70) | 49,335 (67) | 0.02 |
| **Admission source, n (%)** |  |  |  |
| Acute care transfer | 3,200 (14) | 10,275 (21) | <0.0001 |
| Other transfer | 490 (5) | 3,425 (3) |  |
| Non-transfer | 11,720 (81) | 59,350 (76) |  |
| **Cerebral complication, n (%)[d]** | 1,615 (10) | 7,645 (10) | 0.97 |

Abbreviations: ED, emergency department; CCI, Charlson comorbidity index

[a] Drug use was categorized as opioid use (with or without other drugs), cocaine (with or without other non-opioid drugs), other drugs (no opioid or cocaine), and Hepatitis C only. Other drugs include benzodiazepines, amphetamines or stimulants, and hallucinogens.

[b] Other race includes Asian/Pacific Islander, Native American, and 'Other' as categorized by HCUP; races were collapsed due to small cell sizes

[c] Quartile classification of the estimated median household income of residents in the patient's ZIP Code

[d] Includes cerebral abscess, hemorrhage, infarction, and ischemia

**Table 4. Association between drug use, compared to no drug use, on inpatient outcomes.**

| | DU-IE 15,490 (17%) | Non-DU IE 73,505 (83%) | | |
|---|---|---|---|---|
| | N (%) | N (%) | OR (95% CI)[a] | p-value |
| **Discharge disposition** | | | | |
| Routine | 6,010 (39) | 21,270 (29) | 1.0 (ref) | – |
| Home health care | 1,260 (8) | 15,135 (21) | 0.47 (0.39, 0.57) | <0.0001 |
| Transfer, acute care | 2,175 (14) | 6,405 (9) | 1.15 (0.96, 1.37) | 0.12 |
| Transfer, long-term care | 4,865 (31) | 24,090 (33) | 1.60 (1.40, 1.84) | <0.0001 |
| Died | 1,180 (8) | 6,605 (9) | 1.17 (0.95, 1.44) | 0.15 |
| **Inpatient procedure** | | | | |
| Thoracic procedure | 1,600 (10) | 3,310 (5) | 1.23 (0.99, 1.52) | 0.05 |
| Heart valve surgery | 1,955 (13) | 5,530 (8) | 1.17 (0.98, 1.39) | 0.08 |
| Valve replacement | 1,670 (11) | 4,990 (7) | 1.18 (0.98, 1.42) | 0.09 |
| Other | 285 (2) | 540 (1) | 1.11 (0.70, 1.76) | 0.64 |
| | Median (IQR) | Median (IQR) | CIE (95% CI)[a] | p-value |
| **Length of stay, days** | 10 (4–21) | 6 (3–12) | 2.42 (1.59, 3.25) | <0.0001 |
| **Hospital charges, thousands** | 90 (38–201) | 60 (29–135) | -4.20 (-20.56, 12.15) | 0.61 |
| **Thoracic procedure[b], days** | 7 (3–13) | 6 (2–12) | 0.14 (-0.61, 0.89) | 0.71 |
| **Heart valve surgery[b], days** | 7 (3–12) | 5 (1–9) | 2.23 (1.36, 3.10) | <0.0001 |

Abbreviations: OR, odds ratio; CI, confidence interval; IQR; interquartile range; CIE, change in estimate

[a] Adjusted for age, sex, race/ethnicity, primary insurance, median income in the patient's ZIP code, Charlson Comorbidity Index (CCI), pulmonary complications, hospital bed size, teaching status, region, ownership, elective admission, ED status, transfer status, and presence of cerebral complication; complex survey design and weighting were accounted for in analysis; age and CCI were modeled as restricted quadratic splines

[b] Among patients who underwent the procedure only

transfers, and an increased risk of death compared to IE patients without pulmonary complications.

Concordant with these findings, our results demonstrate drug use was the strongest predictor of pulmonary complications. Although this was expected, the downstream effects may have profound implications. In addition to drug use, our study has also identified other predictors of pulmonary complications in IE patients. Predictors associated with increased odds of pulmonary complications included elective admission, acute transfer status, self-pay status, and treatment at an urban institution. Predictors associated with decreased odds of pulmonary complications included older age, higher CCI, NH Black and Hispanic patients, and the presence of cerebral complications. Further study is needed to examine these associations in more detail.

The frequency of DU-IE pulmonary complications remains highest in IE patients between 27–46 years old, similar to the age at highest risk for DU-IE [23]. Hospitalizations for IE have increased in parallel with opioid overdoses with a rising prevalence of injection drug use and has emerged as a rapidly growing public health concern [24]. In this large, national cohort of inpatient hospitalizations, we found that the proportion of DU-IE hospitalizations have remained high and extends the findings of Wurcel et al. and Rudasil et al., which showed a 72% increase between 2000 and 2013 and 190% increase between 2010 and 2015, respectively [25, 26].

In patients with DU-IE, the incidence of septic pulmonary embolism was 27%, while both empyema and lung abscesses were 3%, all of which was higher compared to the non-DU-IE population. These complications also had a more substantial association with poor outcomes (i.e. higher costs, inpatient mortality) when they occurred in a patient with DU-IE compared

to a patient with non-DU-IE. One potential explanation for the higher incidence of pulmonary complications seen in DU-IE compared to non-DU-IE patients may be the higher incidence of right-sided valve disease associated with intravenous drug use [27, 28]. Weber et al. also reported that 27% of right-sided IE patients in their study population suffered from pulmonary complications [29]. Because right-sided IE primarily occur in younger persons, who may have fewer comorbidities and more cardiopulmonary reserve, the diagnosis is often delayed. Additionally, patients who use drugs may delay seeking medical care [30]. This feature may contribute to the longer time to cardiac surgery in right-sided IE compared to left-sided IE patients [29]. In fact, tricuspid IE should be suspected in patients who present with persistent fever, pulmonary symptoms, anemia, and microscopic hematuria—also known as tricuspid syndrome [31]. Future studies are warranted to determine if drug-use is independently associated with pulmonary complications, or whether it is due to differences in disease presentation (left- vs. right-sided IE). We suspect that it is the latter, as right-sided IE is associated with pulmonary complications due to septic pulmonary emboli, resultant abscesses, and pyothorax [32].

Pulmonary complications such as septic pulmonary emboli, empyema, and lung abscess may lead to sepsis and respiratory failure requiring mechanical ventilation and the need for thoracic surgical intervention. Examples include timely treatment of empyema (designated as pyothorax in this study) and ruptured lung abscesses with thoracoscopic versus open pulmonary decortication, pulmonary resection for infarcted or damaged lung, and management of complicated pleural effusions with percutaneous drains and/or thoracostomy tube placement using tPA and DNAse [33–40]. Though not readily identifiable in this database, septic sternoclavicular joint has been associated with intravenous drug use and endocarditis, and we have found this to be not uncommon in our patient population and in need of surgical management [41–43]. Anecdotally, we have observed a notable increase in thoracic surgical consults for septic pulmonary emboli, empyema and lung abscesses in patients admitted with IE at our own institution. Treatment for these patients is complex and complicated, and decisions regarding timing of operation have been based more on anecdotal experience than data due to a paucity of research in this patient population. Between 2015 and 2019 at our institution, 17 (11%) and 34 (21%) of 158 IE patients required pulmonary decortications or percutaneous drains and thoracostomy tube placements, respectively (unpublished data).

These sequelae can ultimately delay definitive cardiac surgical treatment. As noted in the literature, cerebrovascular complications can also impact the timing of cardiac surgery in IE patients, as there is concern that cardiopulmonary bypass (CPB) can exacerbate neurological deficits [44]. For example, systemic anticoagulation needed for CPB may worsen cerebral hemorrhages or convert ischemic infarctions to hemorrhagic infarctions, while hypotension during CPB can aggravate pre-existing neurological injury [44, 45]. However, the current study demonstrated that those with pulmonary complications still waited three days longer for a heart valve operation even after accounting for cerebral complications.

In addition, the timing of surgery could impact the mortality of IE patients, as early cardiac surgical intervention for IE is associated with lower risk of mortality [46]. In our study, we found that, when thoracic and cardiac operations are necessary, performing the operations concurrently was associated with decreased LOS and hospital charges. It is unclear from our study whether concurrent or separate operations represent a surrogate measure of severity of illness. Thus, we can only speculate that the decrease in LOS, and ultimately decreased costs, is attributable to the decrease waiting times between surgeries. Future studies are warranted to determine if an optimal sequence in thoracic procedures and heart valve surgery exists. Current guidelines on the treatment of endocarditis do not address pulmonary complications in the setting of endocarditis nor specify the order in which interventions are performed [47].

The current treatment paradigm for endocarditis includes cardiac surgeons, cardiologists and infectious disease physicians who play a critical role caring for IE patients. However, our study highlights a role for thoracic surgical intervention in patients with IE-PC that is temporally important. We found that patients with pulmonary complications were at increased risk of mortality, particularly DU-IE patients, and optimal thoracic surgical management and guidance could decrease the already high mortality, morbidity and costs associated with IE. However, future study examining the readmission, follow-up, and long-term survival of patients with IE-PC is still needed.

Further, we found that the crude percentage of IE patients without pulmonary complications who died was one percentage point greater than the percentage of IE patients with pulmonary complications who died (9% vs. 8%). However, patients hospitalized with IE and pulmonary complications still had an increased odds of inpatient mortality compared to those without pulmonary involvement after adjustment. This small but reversed association in mortality is likely explained by differences in patient and clinical characteristics. We posit a potential confounding factor that could explain this finding. The Charleson Comorbidity Index (CCI) is a measure of comorbidities at the time of admission and includes diseases such as myocardial infarction, congestive heart failure, peripheral vascular disease, cerebrovascular disease, dementia, chronic pulmonary disease, rheumatologic disease, peptic ulcer disease, liver disease, diabetes mellitus without complications, diabetes complications, hemiplegia/paraplegia, renal disease, metastatic tumor, HIV/AIDS. In our study sample, those with pulmonary complications had a median CCI of 0 (IQR: 0–1), suggesting half of them had no comorbidity, compared to a median CCI of 2 (IQR: 0–3) among those with no pulmonary complications. In addition, 14% of all patients in our sample who died had a CCI of 0. Several studies have shown that hospitalized patients with a higher comorbidity index, suggesting higher number of comorbidities, also tend to have higher mortality rates [48, 49]. Therefore, patients were likely to die of other causes as well. This analysis does not suggest that having pulmonary complications was the cause of death as cause of death is not reported in NIS data. Whereas our analysis accounted for differences in demographics, some clinical and hospital characteristics, and comorbidities between groups, including CCI, we cannot account for all the group-differences reported in Table 1, and it is likely that some residual confounding that we are unable to account for still exists.

## Limitations

Although our study was large and nationally representative, there are several limitations. First, this was an observational study and cannot access causality. We were also limited by the potential for unreported diagnoses (i.e., pulmonary complications and drug use) and potential coding errors within the administrative discharge data; however, this missing data is expected to bias results towards the null (i.e., underestimate the true magnitude of our estimates). This may also explain why a proportion of our patients with no reported thoracic complication underwent a thoracic procedure.

While studies have validated the use of ICD-10 codes for IE, no studies have validated the complications of IE using a "gold standard" medical record review. Additionally, defining drug use as a diagnosis of Hepatitis C in adults ≤50 years old is not a validated method. We pursued this to capture persons with HCV born after the high-prevalence "birth cohort", which ended in 1965, given that contemporary diagnoses of HCV in the US are primarily attributed to injection drug use and we reasoned that HCV infection may serve as a surrogate for injection drug use [50]. This definition has been used previously to best approximate this patient population [16]. Further, specific IE data, such as information on valve location in

non-surgical patients, vegetation size, severity of valvular regurgitation, or cardiac function, as well as the activity status of a patient's drug abuse, identification of the etiologic microorganism, and new or recurrent infections, were not available. Missing data can limit comprehensive inferential analysis and contribute to omitted variable bias. However, our estimates were adjusted for relevant and available variables, which mitigated the effects of omitted variables [51]. Also, because the NIS only captures data during a patient's hospitalization, hospital readmission and survival or follow-up data after patient discharge were not available. Future study using multi-institutional data or the STS National Database is needed for more granular clinical data of patients with IE-PC. And lastly, we were unable to determine the specific indication for surgery, even in patients with identified complications, although we can assume that the majority underwent surgery for complications of interest. Despite these limitations, we believe that our study provides valuable and robust information in the context of pulmonary complications and drug use in the current, rapidly evolving IE population.

## Conclusion

Compared to IE without PC, IE-PC is associated with longer average LOS, higher hospital charges, more frequent post-discharge transfers, and increased odds of death. Patients with DU-IE have a particularly high prevalence of pulmonary complications of IE and was the strongest predictor of pulmonary complications. Moreover, pulmonary complications had a greater impact on poor outcomes in DU-IE patients. Given the significant impact on outcomes we observed in IE-PC patients, future efforts will examine optimal management strategies for patients with IE-PC.

## Supporting information

**S1 File. Contains all the supporting tables.**
(DOCX)

## Author Contributions

**Conceptualization:** Jenny T. Bui, Asher J. Schranz, Jason M. Long.

**Formal analysis:** Paula D. Strassle, Chris B. Agala.

**Methodology:** Paula D. Strassle, Chris B. Agala.

**Software:** Paula D. Strassle.

**Supervision:** Jason M. Long.

**Writing – original draft:** Jenny T. Bui, Asher J. Schranz, Jason M. Long.

**Writing – review & editing:** Jenny T. Bui, Asher J. Schranz, Paula D. Strassle, Chris B. Agala, Gita N. Mody, John S. Ikonomidis, Jason M. Long.

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
