## [Decision Letter · Decision Letter 0]

13 Jan 2021

PONE-D-20-39670

Pulmonary complications on infective endocarditis

PLOS ONE

Dear Dr. Bui,

Thank you for submitting your manuscript to PLOS ONE. After careful consideration, we feel that it has merit but does not fully meet PLOS ONE’s publication criteria as it currently stands. Therefore, we invite you to submit a revised version of the manuscript that addresses the points raised during the review process.

Please revise accordingly.

We look forward to receiving your revised manuscript.

Kind regards,

Academic Editor

PLOS ONE

Journal Requirements:

2. Thank you for providing the date(s) when patient medical information was initially recorded. Please also include the date(s) on which your research team accessed the databases/records to obtain the retrospective data used in your study.

3. Please provide p-values for at least tables 2 and 3 in order to support your claims.

4.Thank you for stating the following in the Funding Section of your manuscript:

"AJS was funded by the National Institute on Drug Abuse (K23DA049946) and the National Institute of Allergy and Infectious Diseases (T32AI070114)."

Reviewers' comments:

Reviewer's Responses to Questions

**Comments to the Author**

1. Is the manuscript technically sound, and do the data support the conclusions?

Reviewer #1: Yes

Reviewer #2: Partly

2. Has the statistical analysis been performed appropriately and rigorously? 

Reviewer #1: N/A

Reviewer #2: No

3. Have the authors made all data underlying the findings in their manuscript fully available?

Reviewer #1: No

Reviewer #2: No

4. Is the manuscript presented in an intelligible fashion and written in standard English?

Reviewer #1: Yes

Reviewer #2: Yes

5. Review Comments to the Author

Reviewer #1: Manuscript PONE-D-20-39670 Thank you for giving us the opportunity to review the manuscript Title: “Pulmonary complications on infective endocarditis” A manuscript in which the author described the short-term outcomes including of infective endocarditis patients with pulmonary complications. the author recommended that pulmonary infarction as a complication of infective endocarditis is associated with worse outcomes. We have some points we would like to refer:

-The site of affected valve not mentioned (right sided or left sided), vegetation size could be helpful for detection and management of pulmonary complication and Predictor prognostic value.

-where are P values for baseline data?

-manuscript should follow the journal instruction regarding arrangement, tables and diagrams.

-alphabetical and grammatical mistakes.

-Death was higher in non-pulmonary complication group 9% vs 8% of pulmonary complication group this is against data in line 278.

Reviewer #2: In Table 1, the rates of those using IV substances and those with HCV infection should be given according to the variables in the column. In addition, the level of significance should be specified by statistically comparing the data in Table 1. In addition, it should be shown statistically whether there is any difference between the two groups, such as other comorbid diseases, central nervous system septic embolism, stroke, symatic embolism to other systems and organs, valvular heart disease, underlying chronic lung disease and immunosuppressive disease.

In the comparison in which mortality was evaluated, were other causes that could affect mortality are controlled?

Wasn't there any AIDS patient among 88,995 patients? What was your purpose in calculating the CCI score? Did you not reach the necessary medical records to calculate this score? Would it be more appropriate to compare the QSofa and SOFA scores between groups instead?

6. PLOS authors have the option to publish the peer review history of their article (what does this mean?). If published, this will include your full peer review and any attached files.

Reviewer #1: No

Reviewer #2: No

---

## [Author Response · Author response to Decision Letter 0]

28 Jan 2021

Academic Editor Comments

1.Please ensure that your manuscript meets PLOS ONE's style requirements, including those for file naming. 

We have revised the manuscript to meet PLOS ONE’s style requirements. 

2.Thank you for providing the date(s) when patient medical information was initially recorded. Please also include the date(s) on which your research team accessed the databases/records to obtain the retrospective data used in your study.

Our research team accessed the NIS database between July 2019 and November 2019 to obtain the retrospective data used in the study. This change is reflected in the “Methods and Materials” section of our paper.

3.Please provide p-values for at least tables 2 and 3 in order to support your claims.

P-values have been added to Tables 2 and 3. 

4.Thank you for stating the following in the Funding Section of your manuscript:

"AJS was funded by the National Institute on Drug Abuse (K23DA049946) and the National Institute of Allergy and Infectious Diseases (T32AI070114)."

Please remove any funding-related text from the manuscript and let us know how you would like to update your Funding Statement. Currently, your Funding Statement reads as follows: "The author(s) received no specific funding for this work."

We have removed any funding-related text from the manuscript. We wish to amend our current financial disclosure statement to state that “AJS was funded by the National Institute on Drug Abuse (K23DA049946) and the National Institute of Allergy and Infectious Diseases (T32AI070114). The funders had no role in study design, data collection and analysis, decision to publish, or preparation of the manuscript.”

5.Please include captions for your Supporting Information files at the end of your manuscript, and update any in-text citations to match accordingly. 

Captions have been included for the Supporting Information files. 

Reviewer Comments #1

1.The site of affected valve and vegetation size could be helpful for detection and management of pulmonary complications and have predictor prognostic value. 

We agree that the site of the affected valve and vegetation size could be helpful for the detection and management of pulmonary complications and add prognostic value. However, the National Inpatient Sample database does not include valve location on non-surgical patients or vegetation size data. This is reflected in the “Limitation” section of the paper.

2.Where are p-values for baseline data?

We did not include p-values for baseline data because it is recommended by the American Statistical Association to exclude p-values on baseline characteristics. With large sample sizes, all baseline data would be statistically significant, and therefore, p-values would not be helpful with interpretation of the data. 

We have provided two articles for your reference here: 

Wasserstein RL, Lazar NA. The ASA Statement on p-Values: Context, Process, and Purpose. The American Statistician. 2016 Apr 2;70(2):129–33.

Wasserstein RL, Schirm AL, Lazar NA. Moving to a World Beyond “p < 0.05.” The American Statistician. 2019 Mar 29;73(sup1):1–19.

3.Manuscript should follow the journal instruction regarding arrangement, tables and diagrams.

The manuscript has been revised to follow journal instruction regarding arrangement, tables, and diagrams. 

4.Alphabetical and grammatical mistakes.

Alphabetical and grammatical mistakes have been revised. 

5.Death was higher in non-pulmonary complication group 9% vs 8% of pulmonary complication group this is against data in line 278.

Although the crude percentage of IE patients with non-pulmonary complications who died was greater than the percentage of IE patients with pulmonary complications who died, patients hospitalized with IE and pulmonary complications still had an increased odds of inpatient mortality (OR 1.81, 95% CI 1.39, 2.35), compared to those without pulmonary involvement after adjustment. This small observed reversed association in mortality is likely explained by differences in patient characteristics (i.e. confounding).

Reviewer Comments #2

1.In Table 1, the rates of those using IV substances and those with HCV infection should be given according to the variables in the column. 

Table 1 has been revised. We have also added footnotes to clarify how IV drug use was assigned and analyzed in 

the table. 

2.The level of significance should be specified by statistically comparing the data in Table 1. In addition, it should be shown statistically whether there is any difference between the two groups, such as other comorbid diseases, central nervous system septic embolism, stroke, symptomatic embolism to other systems and organs, valvular heart disease, underlying chronic lung disease and immunosuppressive disease.

As previously mentioned above, we did not include p-values for the baseline data in Table 1 due to the current recommendations by the American Statistical Association. Further, given that this is a relatively younger and ‘healthier’ patient population, we chose to use a composite score of the Charlson Comorbidity Index rather than including specific comorbidities due to the likely low prevalence of each individual comorbidity. 

3.In the comparison in which mortality was evaluated, were other causes that could affect mortality controlled?

Yes, we controlled for several patient characteristics when assessing all-cause in-patient mortality. Specifically, we adjusted for age, sex, race/ethnicity, primary insurance, median income in the patient’s ZIP code, Charlson Comorbidity Index, IV drug use, hospital bed size, teaching status, region, ownership, elective admission, ED status, transfer status, and presence of cerebral complication. This is noted in our Methods section and included in the footnotes of Table 2 and 3. Although, it is likely that there may be some residual confounding (e.g. frailty) that we are unable to control and account for. We have added this to our Limitations section.

We did not and cannot assess specific causes of death in this dataset.

4.Wasn't there any AIDS patient among 88,995 patients? 

Thank you for highlighting that HIV is a co-occurring concern, particularly in persons who inject drugs. In this study, we did not specifically examine HIV or AIDS as a comorbidity. AIDS is included in the Charlson Comorbidity Index (CCI), so the comorbidity conferred by AIDS is accounted for in that scoring. We opted not to specifically highlight the number of hospitalizations coded for AIDS, or other specific comorbidities, since this would introduce substantial complexity, given that many multisystem comorbidities may impact outcomes in patients with endocarditis or those undergoing surgery. We instead relied on the CCI as a summative measure. 

5.What was your purpose in calculating the CCI score? Did you not reach the necessary medical records to calculate this score? Would it be more appropriate to compare the QSofa and SOFA scores between groups instead?

We opted to not highlight specific comorbidities given the likely low prevalence of these comorbidities in our patient population and the potential that many multisystem comorbidities may impact outcomes in patients with endocarditis or those undergoing surgery. We instead relied on the CCI as a summative measure. QSOFA and SOFA scores cannot be calculated using the data available in the National Inpatient Sample database.

---

## [Decision Letter · Decision Letter 1]

15 Feb 2021

PONE-D-20-39670R1

Pulmonary complications observed in patients with infective endocarditis

PLOS ONE

Dear Dr. Bui,

Thank you for submitting your manuscript to PLOS ONE. After careful consideration, we feel that it has merit but does not fully meet PLOS ONE’s publication criteria as it currently stands. Therefore, we invite you to submit a revised version of the manuscript that addresses the points raised during the review process.

Please revise accordingly.

We look forward to receiving your revised manuscript.

Kind regards,

Academic Editor

PLOS ONE

Reviewers' comments:

Reviewer's Responses to Questions

**Comments to the Author**

1. If the authors have adequately addressed your comments raised in a previous round of review and you feel that this manuscript is now acceptable for publication, you may indicate that here to bypass the “Comments to the Author” section, enter your conflict of interest statement in the “Confidential to Editor” section, and submit your "Accept" recommendation.

Reviewer #1: (No Response)

Reviewer #2: All comments have been addressed

2. Is the manuscript technically sound, and do the data support the conclusions?

Reviewer #1: No

Reviewer #2: Yes

3. Has the statistical analysis been performed appropriately and rigorously? 

Reviewer #1: No

Reviewer #2: Yes

4. Have the authors made all data underlying the findings in their manuscript fully available?

Reviewer #1: No

Reviewer #2: Yes

5. Is the manuscript presented in an intelligible fashion and written in standard English?

Reviewer #1: No

Reviewer #2: Yes

6. Review Comments to the Author

Reviewer #1: Manuscript PONE-D-20-39670R1 Thank you for giving us the opportunity to review the manuscript Title: “Pulmonary complications on infective endocarditis” A manuscript in which the author described the short-term outcomes including of infective endocarditis patients with pulmonary complications. the author recommended that pulmonary infarction as a complication of infective endocarditis is associated with worse outcomes. unfortunately the author did not address any of the previously mentioned recommendations with clear explanation other than no available data.

Reviewer #2: It would be much more useful and instructive if this study included the affected heart valve, culture results and causative microorganism. Author should put down this inadequacies in limitation part.

7. PLOS authors have the option to publish the peer review history of their article (what does this mean?). If published, this will include your full peer review and any attached files.

Reviewer #1: No

Reviewer #2: **Yes: **Ali Acar

---

## [Author Response · Author response to Decision Letter 1]

8 Mar 2021

Reviewer Comments #1

1.Unfortunately, the author did not address any of the previously mentioned recommendations with clear explanation other than no available data.

We have included the previously requested baseline p-values in Table 1. Given the statistically significant demographic differences seen in Table 1 between the pulmonary complication and no pulmonary complication groups, we conducted secondary inferential analyses to gauge potential effects of a large sample size on the multivariate estimates of group comparisons. We assessed crude associations between outcomes and hospital regions as this was the only non-significant comparison in Table 1. The analysis showed some regional variations in all outcomes except heart valve surgery, even though the presence of pulmonary complications did not vary by hospital region (S3 Table). These results demonstrate that the differences seen in our outcomes of interest, such as length of stay, hospital charges, and discharge disposition (including inpatient mortality), are not only due to the differences seen in baseline characteristics of our study population. 

As previously suggested, we have proofread the manuscript for any grammatical and spelling mistakes and have included that the reviewer’s suggested variables (i.e. site of affected valve and vegetation size) are not available to be collected and analyzed in the National Inpatient Sample database in our “Limitations” section. 

Reviewer Comments #2 

1.It would be much more useful and instructive if this study included the affected heart valve, culture results and causative microorganism. Author should put down these inadequacies in limitation part.

We agree that the site of the affected valve, culture results, and causative microorganism would have been useful in our study analysis. However, the National Inpatient Sample database does not include valve location on non-surgical patients, culture results, or causative microorganism. This is now reflected in the “Limitations” section of the paper.

---

## [Decision Letter · Decision Letter 2]

23 Mar 2021

PONE-D-20-39670R2

Pulmonary complications observed in patients with infective endocarditis

PLOS ONE

Dear Dr. Bui,

Thank you for submitting your manuscript to PLOS ONE. After careful consideration, we feel that it has merit but does not fully meet PLOS ONE’s publication criteria as it currently stands. Therefore, we invite you to submit a revised version of the manuscript that addresses the points raised during the review process.

Please revise accordingly.

We look forward to receiving your revised manuscript.

Kind regards,

Academic Editor

PLOS ONE

Reviewers' comments:

Reviewer's Responses to Questions

**Comments to the Author**

1. If the authors have adequately addressed your comments raised in a previous round of review and you feel that this manuscript is now acceptable for publication, you may indicate that here to bypass the “Comments to the Author” section, enter your conflict of interest statement in the “Confidential to Editor” section, and submit your "Accept" recommendation.

Reviewer #3: All comments have been addressed

Reviewer #4: (No Response)

Reviewer #5: All comments have been addressed

2. Is the manuscript technically sound, and do the data support the conclusions?

Reviewer #3: Partly

Reviewer #4: Yes

Reviewer #5: Yes

3. Has the statistical analysis been performed appropriately and rigorously? 

Reviewer #3: Yes

Reviewer #4: Yes

Reviewer #5: Yes

4. Have the authors made all data underlying the findings in their manuscript fully available?

Reviewer #3: Yes

Reviewer #4: No

Reviewer #5: Yes

5. Is the manuscript presented in an intelligible fashion and written in standard English?

Reviewer #3: Yes

Reviewer #4: Yes

Reviewer #5: Yes

6. Review Comments to the Author

Reviewer #3: I suggests to change the aim and also the title of this paper. Most relevant conlcusion of this paper is the high percentage of pulmonary complications observed in patients with endocarditis related to drug abuse. The greater impact on outcome of pulmonary complications in patients with endocarditis is expected and obvious and no data are available on major determints of pulmonary complications

Reviewer #4: We have made a review for Manuscript Number PONE-D-20-39670R2; entitled: Pulmonary complications observed in patients with infective endocarditis. We have some comments:

1- The background is not coherent and still does not demonstrate strong rationale supported by proper literature.

2- In the methodology, the authors should describe the inclusion and exclusion criteria in a clear manner. Moreover, better to define primary and secondary outcome of your search.

3- The authors have selected the year of 2016 which was the first year to use ICD 10. This may mean higher human error.

4- No details on survival or follow up. Data were collected from administrative electronic archive and lack important clinical data.

5- The large sample size from electronic record has its own limitations. The authors should defend more regarding this point.

6- Search and analyse important data like hospital readmission within 90 days after discharge.

Regards

Reviewer #5: Dear The authors

Thank you for writing this manuscript and I am convinced with your responses to the reviewers comments

I personally believe that the idea covered by this manuscript is an excellent and is an important addition to the field of cardiac surgery

I have no concerns about it

Thank you

7. PLOS authors have the option to publish the peer review history of their article (what does this mean?). If published, this will include your full peer review and any attached files.

Reviewer #3: **Yes: **luciano agati

Reviewer #4: **Yes: **Rami Riziq Yousef Abumuaileq

Reviewer #5: **Yes: **salah eldien Altarabsheh

---

## [Author Response · Author response to Decision Letter 2]

4 Apr 2021

Reviewer Comments #3

1. I suggest changing the aim and also the title of this paper. Most relevant conclusion of this paper is the high percentage of pulmonary complications observed in patients with endocarditis related to drug abuse. The greater impact on outcome of pulmonary complications in patients with endocarditis is expected and obvious and no data are available on major determinants of pulmonary complications.

Thank you for your suggestions. We have changed the title of our paper to, “Pulmonary complications observed in patients with infective endocarditis with and without injection drug use: an analysis of the National Inpatient Sample”. However, we believe that the aims of our study were appropriate for our research question. Although, the greater impact on outcome of pulmonary complications in patients with IE may be expected, there is currently no literature with statistical data that supports this hypothesis. Our paper provides this evidence for both IE patients with and without drug use. 

Reviewer Comments #4

1. The background is not coherent and still does not demonstrate strong rationale supported by proper literature.

We have revised the background with stronger rationale, including adding new references and figures. 

2. In the methodology, the authors should describe the inclusion and exclusion criteria in a clear manner. Moreover, better to define primary and secondary outcome of your search.

All hospitalizations of adult (≥18 years old) patients diagnosed with IE were identified using ICD-10-CM codes and included in the study population. Patients discharged against medical advice or with unknown disposition were excluded from the study population since their course of care and clinical outcomes could not be reliably assessed.

The primary outcome of interest was inpatient mortality in IE-PC patients (with and without drug use). The secondary outcomes of interest included undergoing a thoracic and/or heart valve procedure, discharge disposition, LOS, total hospital charges, and time to thoracic or heart valve procedure (among those who underwent procedure only). 

3. The authors have selected the year of 2016 which was the first year to use ICD 10. This may mean higher human error.

Yes, we agree with your comment and have mentioned the potential for coding errors with using an administrative discharge database in the “Limitations” section. However, this missing data is expected to bias results towards the null (i.e. underestimate the true magnitude of our estimates).

4. No details on survival or follow up. Data were collected from administrative electronic archive and lack important clinical data.

We recognize that this is a limitation of a large administrative database. The National Inpatient Sample database only includes data during a patient’s hospitalization. Therefore, details on survival or follow-up after discharge are not captured in the database. This now reflected in the “Limitations” section of our manuscript. However, this is an excellent topic for future study using multi-institutional data or the STS National Database that allows for more granular data collection. 

5. The large sample size from electronic record has its own limitations. The authors should defend more regarding this point.

We agree that a large sample size has its own limitations. Of note, it can be likely to find a difference by chance. However, for that reason, we performed a secondary inferential analysis to assess the crude association between outcomes and hospital regions. As mentioned in the “Methodology” section, the analysis was performed to gauge the potential effect of a large sample size. The analysis confirmed that our findings were not by chance but were in fact real effects as seen in the “Results” section and Supplemental Table 3. We have also adhered to the guidelines suggested by JAMA Surgery for improving the quality of database research. 

Haider AH, Bilimoria KY, Kibbe MR. A Checklist to Elevate the Science of Surgical Database Research. JAMA Surg. 2018;153(6):505. doi:10.1001/jamasurg.2018.0628

6. Search and analyze important data like hospital readmission within 90 days after discharge.

While we agree that hospital readmission data is important, the National Inpatient Sample database does not include hospital readmission data. This is now reflected in the “Limitations” section of our manuscript. This is also an excellent topic for future study using multi-institutional data or the STS National Database that allows for more granular data collection.

---

## [Decision Letter · Decision Letter 3]

11 Apr 2021

PONE-D-20-39670R3

Pulmonary complications observed in patients with infective endocarditis with and without injection drug use: an analysis of the National Inpatient Sample

PLOS ONE

Dear Dr. Bui,

Thank you for submitting your manuscript to PLOS ONE. After careful consideration, we feel that it has merit but does not fully meet PLOS ONE’s publication criteria as it currently stands. Therefore, we invite you to submit a revised version of the manuscript that addresses the points raised during the review process.

Please address the reviewer Dr. Agati's questions and revise accordingly.

We look forward to receiving your revised manuscript.

Kind regards,

Academic Editor

PLOS ONE

Journal Requirements:

Reviewers' comments:

Reviewer's Responses to Questions

**Comments to the Author**

1. If the authors have adequately addressed your comments raised in a previous round of review and you feel that this manuscript is now acceptable for publication, you may indicate that here to bypass the “Comments to the Author” section, enter your conflict of interest statement in the “Confidential to Editor” section, and submit your "Accept" recommendation.

Reviewer #3: (No Response)

Reviewer #5: All comments have been addressed

2. Is the manuscript technically sound, and do the data support the conclusions?

Reviewer #3: Partly

Reviewer #5: Yes

3. Has the statistical analysis been performed appropriately and rigorously? 

Reviewer #3: Yes

Reviewer #5: Yes

4. Have the authors made all data underlying the findings in their manuscript fully available?

Reviewer #3: Yes

Reviewer #5: Yes

5. Is the manuscript presented in an intelligible fashion and written in standard English?

Reviewer #3: Yes

Reviewer #5: Yes

6. Review Comments to the Author

Reviewer #3: My major comment was not addressed. Patients with IE and pulmonary complications have worse outcome and this is obvious, however, more interesting, major determinants of pulmonary complications in patients with IE were not studied

Reviewer #5: Dear the Authors

Thank you again for taking care of all the review comments

I have nom concerns about this manuscript

7. PLOS authors have the option to publish the peer review history of their article (what does this mean?). If published, this will include your full peer review and any attached files.

Reviewer #3: **Yes: **luciano agati

Reviewer #5: **Yes: **Salah Altarabsheh

---

## [Author Response · Author response to Decision Letter 3]

11 May 2021

Dear editor and reviewers, 

We thank the academic editor and reviewers for taking the time to re-review our paper, and we appreciate their comments. 

We have attached the comments and our responses below: 

Editor Comments: 

The reference list has been reviewed for accuracy. We did not make any changes to the reference list or cite any retracted articles.

Reviewer #3 Comments: 

1. My major comment was not addressed. Patients with IE and pulmonary complications have worse outcomes, and this is obvious, however, more interesting, major determinants of pulmonary complications in patients with IE were not studied. 

We have added additional analyses examining the predictors of pulmonary complications in patients with IE. It was confirmed in our analysis that DU-IE patients had the highest odds of pulmonary complications. Other predictors of pulmonary complications included: elective admission to the hospital, acute transfer status, self-paying status, and treatment at an urban institution. Predictors associated with decreased odds of pulmonary complications included older age, higher CCI, NH Black and Hispanic patients, and the presence of cerebral complications. This is now reflected in our Aims, Methods, Results, and Discussion of our paper.

---

## [Decision Letter · Decision Letter 4]

17 May 2021

PONE-D-20-39670R4

Pulmonary complications observed in patients with infective endocarditis with and without injection drug use: an analysis of the National Inpatient Sample

PLOS ONE

Dear Dr. Bui,

Thank you for submitting your manuscript to PLOS ONE. After careful consideration, we feel that it has merit but does not fully meet PLOS ONE’s publication criteria as it currently stands. Therefore, we invite you to submit a revised version of the manuscript that addresses the points raised during the review process.

Please revise accordingly.

We look forward to receiving your revised manuscript.

Kind regards,

Academic Editor

PLOS ONE

Reviewers' comments:

Reviewer's Responses to Questions

**Comments to the Author**

1. If the authors have adequately addressed your comments raised in a previous round of review and you feel that this manuscript is now acceptable for publication, you may indicate that here to bypass the “Comments to the Author” section, enter your conflict of interest statement in the “Confidential to Editor” section, and submit your "Accept" recommendation.

Reviewer #3: All comments have been addressed

Reviewer #5: All comments have been addressed

Reviewer #6: (No Response)

2. Is the manuscript technically sound, and do the data support the conclusions?

Reviewer #3: Yes

Reviewer #5: Yes

Reviewer #6: Partly

3. Has the statistical analysis been performed appropriately and rigorously? 

Reviewer #3: Yes

Reviewer #5: Yes

Reviewer #6: Yes

4. Have the authors made all data underlying the findings in their manuscript fully available?

Reviewer #3: Yes

Reviewer #5: Yes

Reviewer #6: Yes

5. Is the manuscript presented in an intelligible fashion and written in standard English?

Reviewer #3: Yes

Reviewer #5: Yes

Reviewer #6: Yes

6. Review Comments to the Author

Reviewer #3: (No Response)

Reviewer #5: Thank you for taking care of the reviewers comments

I think they are satisfactory and the manuscript looks better

Reviewer #6: In this manuscript, the authors studied the impact of pulmonary complications on the outcome of infective endocarditis using NIS data in 2016. The authors identified that the pulmonary complications were associated with longer length of hospital stay and increased odds of inpatient mortality. The reviewer finds the study's focus interesting, but has some concerns about the results.

1. According to the Table 3, it seems that the mortality is significantly higher in the patients without pulmonary complication (8% vs 9%, OR 1.81, p<0.0001). This result is completely opposite to the interpretation in the manuscript, so the authors need to clarify this point.

2. The authors divided the patients into groups named “Pulmonary complication” and “No complication” in Table 1, 2 and 3. However, the reviewer thinks “No complication” is a misleading term. “No pulmonary complication” or “Without pulmonary complication” should be appropriate.

3. There are significant overlap in the contents between Table 1 and Table 2. The reviewer thinks it is better to put aside one of which to the supplementary data.

4. Table 4 only shows the patients outcomes. The comparison of patient demographics and characteristics according to IV drug use should be also presented in a Table format.

5. Because only 7% of the whole population has pulmonary complication, the absolute number of drug users is higher in patients without pulmonary complications. Therefore, the number and percentage of each group should be listed in the first column of Table 4 (as in Table 1 and Table 2).

7. PLOS authors have the option to publish the peer review history of their article (what does this mean?). If published, this will include your full peer review and any attached files.

Reviewer #3: **Yes: **luciano Agati

Reviewer #5: **Yes: **Salah Altarabsheh

Reviewer #6: No

---

## [Author Response · Author response to Decision Letter 4]

8 Jun 2021

Dear editor and reviewers, 

We thank the academic editor and reviewers for taking the time to re-review our paper, and we appreciate their comments. We have attached the comments and our responses below: 

Reviewer #6: 

1. According to the Table 3, it seems that the mortality is significantly higher in the patients without pulmonary complication (8% vs 9%, OR 1.81, p<0.0001). This result is completely opposite to the interpretation in the manuscript, so the authors need to clarify this point.

Although the crude percentage of IE patients with non-pulmonary complications who died was greater than the percentage of IE patients with pulmonary complications who died, patients hospitalized with IE and pulmonary complications still had an increased odds of inpatient mortality (OR 1.81, 95% CI 1.39, 2.35), compared to those without pulmonary involvement after adjustment. This small observed reversed association in mortality is likely explained by differences in patient characteristics (i.e. confounding). This point has been clarified in the Results and Limitations section. 

2.The authors divided the patients into groups named “Pulmonary complication” and “No complication” in Table 1, 2 and 3. However, the reviewer thinks “No complication” is a misleading term. “No pulmonary complication” or “Without pulmonary complication” should be appropriate.

Thank you for the suggestion. We have changed the term to “No Pulmonary Complication” in Table 1, 2, and Supplemental Table 3. 

3.There are significant overlap in the contents between Table 1 and Table 2. The reviewer thinks it is better to put aside one of which to the supplementary data.

Table 2 have been moved to the supplementary data as S3 Table. 

4.Table 4 only shows the patients outcomes. The comparison of patient demographics and characteristics according to IV drug use should be also presented in a Table format.

We have added a Table 3 that includes patient demographics and hospital characteristics among patients with and without drug use. 

5. Because only 7% of the whole population has pulmonary complication, the absolute number of drug users is higher in patients without pulmonary complications. Therefore, the number and percentage of each group should be listed in the first column of Table 4 (as in Table 1 and Table 2).

We were a bit confused on this suggestion but tried our best to interpret your suggestion. We have added the total number and percentage of DU-IE and non-DU-IE patients to the first row in Table 4. We will be happy to revise further if this was not your intended suggestion.

---

## [Decision Letter · Decision Letter 5]

15 Jun 2021

PONE-D-20-39670R5

Pulmonary complications observed in patients with infective endocarditis with and without injection drug use: an analysis of the National Inpatient Sample

PLOS ONE

Dear Dr. Bui,

Thank you for submitting your manuscript to PLOS ONE. After careful consideration, we feel that it has merit but does not fully meet PLOS ONE’s publication criteria as it currently stands. Therefore, we invite you to submit a revised version of the manuscript that addresses the points raised during the review process.

Please address the issues and revise accordingly. 

We look forward to receiving your revised manuscript.

Kind regards,

Academic Editor

PLOS ONE

Reviewers' comments:

Reviewer's Responses to Questions

**Comments to the Author**

1. If the authors have adequately addressed your comments raised in a previous round of review and you feel that this manuscript is now acceptable for publication, you may indicate that here to bypass the “Comments to the Author” section, enter your conflict of interest statement in the “Confidential to Editor” section, and submit your "Accept" recommendation.

Reviewer #5: All comments have been addressed

Reviewer #6: (No Response)

2. Is the manuscript technically sound, and do the data support the conclusions?

Reviewer #5: Yes

Reviewer #6: Partly

3. Has the statistical analysis been performed appropriately and rigorously? 

Reviewer #5: Yes

Reviewer #6: I Don't Know

4. Have the authors made all data underlying the findings in their manuscript fully available?

Reviewer #5: Yes

Reviewer #6: No

5. Is the manuscript presented in an intelligible fashion and written in standard English?

Reviewer #5: Yes

Reviewer #6: Yes

6. Review Comments to the Author

Reviewer #5: Dear the authors

Thank you for your efforts in taking considerations of all the reviewers comments

I think this manuscript stands in good shape

Reviewer #6: The authors addressed the concerns raised by the reviewer except for the major problem #1; the authors provide an explanation for the reversed association in mortality as a response to the reviewer, but it is not reflected in the manuscript, although they stated that “This point has been clarified in the Results and Limitations section”.

The fact that "patients with pulmonary complications have a statistically significantly lower crude mortality rate, but when adjusted for factors such as age and gender, the results are reversed" is extremely important and should not be dismissed out of hand.

Looking at Table 1, patients with pulmonary complications are younger and have lower rates of cerebrovascular complications, so it is unlikely that the odds ratio would be reversed to 1.8 even after adjustment for clinical factors. It is necessary to make an effort to identify which confounders are influencing the results, and it is the minimum obligation of scientists to state this in discussions and limitations. Otherwise, it is suspected that the method of analysis and interpretation may have been incorrect.

The reviewer is well aware that 6 times of revision is exhausting, but believes that the problem listed above should be the final barrier to publication.

7. PLOS authors have the option to publish the peer review history of their article (what does this mean?). If published, this will include your full peer review and any attached files.

Reviewer #5: **Yes: **salah eldien Altarabsheh

Reviewer #6: No

---

## [Author Response · Author response to Decision Letter 5]

26 Jul 2021

Dear editor and reviewers, 

We thank the academic editor and reviewers for taking the time to re-review our paper, and we appreciate their comments. 

We have attached the comments and our responses below: 

Reviewer #6: 

1.The authors addressed the concerns raised by the reviewer except for the major problem #1; the authors provide an explanation for the reversed association in mortality as a response to the reviewer, but it is not reflected in the manuscript, although they stated that “This point has been clarified in the Results and Limitations section”.

The fact that "patients with pulmonary complications have a statistically significantly lower crude mortality rate, but when adjusted for factors such as age and gender, the results are reversed" is extremely important and should not be dismissed out of hand. Looking at Table 1, patients with pulmonary complications are younger and have lower rates of cerebrovascular complications, so it is unlikely that the odds ratio would be reversed to 1.8 even after adjustment for clinical factors. It is necessary to make an effort to identify which confounders are influencing the results, and it is the minimum obligation of scientists to state this in discussions and limitations. Otherwise, it is suspected that the method of analysis and interpretation may have been incorrect.

As previously mentioned, although the crude percentage of IE patients without pulmonary complications who died was greater than the percentage of IE patients with pulmonary complications who died, patients hospitalized with IE and pulmonary complications still had an increased odds of inpatient mortality (OR 1.81, 95% CI 1.39, 2.35), compared to those without pulmonary involvement after adjustment. In fact, our additional analysis shows that the odds ratio for mortality compared to routine discharge disposition was lower among patients with no pulmonary complications compared to those with pulmonary complications (OR: 0.878; 95% CI: 0.703 ,1.095), in line with crude frequencies of 8% (pulmonary omplications) vs. 9% (without pulmonary complications) in Table 2. This small observed reversed association in mortality is likely explained by differences in patient and clinical characteristics which were included in the adjusted model and produced the reported adjusted odds ratio of 1.81. One potential confounding factor that could explain this finding includes: 

1)The Charleson Comorbidity Index (CCI) is a measure of comorbidities at the time of admission and includes diseases such as myocardial infarction, congestive heart failure, peripheral vascular disease, cerebrovascular disease, dementia, chronic pulmonary disease, rheumatologic disease, peptic ulcer disease, liver disease, diabetes mellitus without complications, diabetes complications, hemiplegia/paraplegia, renal disease, metastatic tumor, HIV/AIDS. In Table 1, we observe that those with pulmonary complications had a median CCI of 0 (IQR:0-1), suggesting half of them had no comorbidity, compared to a median CCI of 2 (IQR:0-3) among those with no pulmonary complications. Those with a higher number of comorbidities also tend to have higher mortality rates. Therefore, patients were likely to die of other causes as well. Additional analysis shows that 14% of all patients who died had a CCI value of 0, the rest had a higher CCI value. Overall, this analysis does not suggest that having pulmonary complications was the cause of death as cause of death is not reported in NIS data. It is also possible that some of this difference could be accounted for by omitted variables, which were also unavailable in the database—one of the limitations of the NIS database (please see limitations of NIS in McGinigle et al. 2021 (https://www.sciencedirect.com/science/article/pii/S0741521420325064?via%3Dihub))

While we did account for differences in demographics, some clinical and hospital characteristics, and comorbidities between groups, including CCI, we cannot account for all the differences in Table 1, and it is likely that some residual confounding that we are unable to account for still exists. Please see this text in the Discussion section on page 16, and in Limitations section on page 17, with added relevant citations.

---

## [Decision Letter · Decision Letter 6]

16 Aug 2021

Pulmonary complications observed in patients with infective endocarditis with and without injection drug use: an analysis of the National Inpatient Sample

PONE-D-20-39670R6

Dear Dr. Bui,

We’re pleased to inform you that your manuscript has been judged scientifically suitable for publication and will be formally accepted for publication once it meets all outstanding technical requirements.

Kind regards,

Academic Editor

PLOS ONE

Additional Editor Comments (optional):

Reviewers' comments:

Reviewer's Responses to Questions

**Comments to the Author**

1. If the authors have adequately addressed your comments raised in a previous round of review and you feel that this manuscript is now acceptable for publication, you may indicate that here to bypass the “Comments to the Author” section, enter your conflict of interest statement in the “Confidential to Editor” section, and submit your "Accept" recommendation.

Reviewer #5: All comments have been addressed

Reviewer #6: All comments have been addressed

2. Is the manuscript technically sound, and do the data support the conclusions?

Reviewer #5: Yes

Reviewer #6: Yes

3. Has the statistical analysis been performed appropriately and rigorously? 

Reviewer #5: Yes

Reviewer #6: Yes

4. Have the authors made all data underlying the findings in their manuscript fully available?

Reviewer #5: Yes

Reviewer #6: Yes

5. Is the manuscript presented in an intelligible fashion and written in standard English?

Reviewer #5: Yes

Reviewer #6: Yes

6. Review Comments to the Author

Reviewer #5: Dear the authors

Thank you very much for taking consideration for all the reviewers comments

I have no concerns

Reviewer #6: The authors addressed the concern raised by the reviewer at the previous round of review.

The reviewer feels now the manuscript meets the publication criteria.

7. PLOS authors have the option to publish the peer review history of their article (what does this mean?). If published, this will include your full peer review and any attached files.

Reviewer #5: **Yes: **Salah Eldien Altarabsheh

Reviewer #6: No

---

## [Editor Report · Acceptance letter]

27 Aug 2021

PONE-D-20-39670R6 

Pulmonary complications observed in patients with infective endocarditis with and without injection drug use: an analysis of the National Inpatient Sample 

Dear Dr. Bui:

I'm pleased to inform you that your manuscript has been deemed suitable for publication in PLOS ONE. Congratulations! Your manuscript is now with our production department. 

Kind regards, 

on behalf of

Dr. Robert Jeenchen Chen 

Academic Editor

PLOS ONE